# Effect of Acid–Base Modified Biochar on Chlortetracycline Adsorption by Purple Soil



**Zhifeng Liu** [1,2,3], **Xun Fang** [1], **Lingyuan Chen** [4], **Bo Tang** [1,2,3], **Fengmin Song** [1,2,3] **and Wenbin Li** [4,*]

1   School of Chemistry and Environmental Science, Shaanxi University of Technology, Hanzhong 723001, China; lzhifeng2005@126.com (Z.L.); fx187910@163.com (X.F.); czhatb@163.com (B.T.); sfm3297@126.com (F.S.)
2   State Key Laboratory of Qinba Bio-Resource and Ecological Environment, Hanzhong 723001, China
3   Shaanxi Province Key Laboratory of Catalytic Foundation and Application, Hanzhong 723001, China
4   College of Environmental Science and Engineering, China West Normal University, Nanchong 637009, China; lychen_cwnu@163.com
*   Correspondence: lwb062@cwnu.edu.cn

**Abstract:** We used three purple soil (Hechuan, Jialing, and Cangxi) samples from the Jialing River basin as the research objects and added different proportions of an acid–base modified *Alternanthera philoxeroides* biochar ($C_m$) to the purple soil to study the effect of $C_m$ on the adsorption of chlortetracycline (CTC) in the purple soil. The results indicated the following: (1) At 30 °C and pH = 6, the soil adsorption capacity increased with an increasing initial concentration of CTC. The maximum adsorption amount of CTC for each tested sample was in the range of 2054.63–3631.21 mg/kg, and the adsorption capacity in different $C_m$ amended soils was ranked in the order of 10% $C_m$ > 5% $C_m$ > 2% $C_m$ > CK. The adsorption capacity of CTC increased with an increase in the proportion of $C_m$. Furthermore, under the same addition ratio of $C_m$, Hechuan soil was found to have a better adsorption effect for CTC than Jialing and Cangxi soil. (2) The Langmuir model was the most suitable for fitting the adsorption behavior of CTC on different purple soils, and the fitting coefficients were all greater than 0.9, indicating that the adsorption of CTC on each soil sample occurred via monolayer adsorption. The thermodynamic experiment results showed that an increase in temperature was beneficial to the process of CTC adsorption, which was a spontaneous, endothermic, and entropy-adding process. (3) At pH = 6, the ionic strength ranged from 0.01 to 0.5 mol/L and the adsorption capacity of CTC of the soil samples decreased with an increase in ionic strength. In the range of pH 2–10, the adsorption capacity of CTC in all the soil samples decreased with an increase in pH. The inhibition capacity of CTC in the soil samples under acidic conditions was notably higher than that under alkaline conditions.

**Keywords:** *Alternanthera philoxeroides*; biochar; modification; chlortetracycline; adsorption capacity; Jialing River

## 1. Introduction

As one of the great successes of science, antibiotics not only benefit society but also cause environmental pollution that cannot be ignored [1]. The inappropriate and irrational use of antibiotics is the root cause of antibiotic resistance [2]. Studies have shown that the extensive use of antibiotics in animal husbandry is an important source of antibiotic pollution in the soil environment [3]. According to the statistics of the Chemical Industry Association and Pharmaceutical Industry Association in 2005, China produces approximately 210,000 tons of raw materials for antibiotics every year, of which 97,000 tons (46.1% of the total annual output) is used in animal husbandry; and in the United States, artificial antibiotics used as feed supplements also exceeded 50% [4,5]. According to the survey, in 1996, feed with antibiotics accounted for 45.8% of all feed additives, and approximately 70% of antibiotics were used in animal husbandry [6,7]. However, only a small part of antibiotics is absorbed by livestock, and the vast majority of antibiotics will enter the environment as

feces, thereby resulting in the generation of resistant microorganisms and genes resistant to antibiotics in the soil and water environment [8,9]. Tetracycline antibiotics are widely used by livestock farms to prevent the outbreak and spread of diseases [10]. However, the stomach and intestines have trouble absorbing antibiotics during digestion. As such, antibiotics enters the soil and water environment through animal feed additives, manure, irrigation, and other routes [11]. Previous studies showed that 80–95% of antibiotics added to feed are excreted through feces and urine, thereby resulting in a maximum antibiotics concentration of 563.8 mg/kg in feces [12,13]. Wang et al. [14] studied the status of antibiotic residues in the manure, feed, and soil of livestock and poultry in the Erhai-Lake Basin and found that the content of antibiotics in pig feed was the highest, and 90% of the antibiotics residues were detected in manure. Zhang et al. [15] measured the main chemical composition of typical large-scale livestock and poultry manure from seven provinces, cities, and autonomous regions in China, and the results showed that the antibiotics content in pig and chicken manure was 3.57 mg/kg and 1.39 mg/kg, respectively.

So far, the methods for removing antibiotics from soil and water mainly include ozonation, adsorption, photolysis, photocatalysis, Fenton, microbial treatment, electrolysis, and so on [16,17]. Among the abovementioned methods, the adsorption method is considered the most effective because of its simple operation and high selectivity [18,19]. Biochar is a carbon-rich solid that can be pyrolyzed from various types of biomass wastes [20]. As an eco-friendly adsorbent, it is often used in organically contaminated soil and water [21]. The modification of biochar changes the specific surface area, voids, and functional groups of biochar through some chemical and physical methods to improve the adsorption capacity of biochar for various pollutants. It is a practical and environmentally friendly method [22,23]. Jing et al. [24] modified rice husk biochar with methanol, and the results showed that the adsorption capacity of modified biochar increased by 45.6% within 12 h and 17.20% within equilibrium time. Liang et al. [25] synthesized a new type of modified biochar (MBC) by loading magnetic manganese iron oxide nanoparticles on pine biochar (Mn-Fe-BC). The results showed that the maximum adsorption capacity of Mn-Fe-BC for tetracycline reached 177.71 mg/g, and toxicity studies showed that Mn-Fe-BC is not harmful to the environment. The adsorption performance of biochar can be greatly improved after various modification treatments. If modified biochar is added to the actual contaminated soil sample as a modified material, the adsorption capacity of soil for antibiotics could be greatly improved. Meanwhile, the biochar material improved the physical and chemical properties of soil simultaneously, showing a good application prospect.

*Alternanthera philoxeroides* is an exotic weed that grows quickly and has great stress resistance. It is adapted to aquatic and terrestrial environments and breeds and spreads in many areas of our country, thereby causing serious damage [26]. As a result of its large spread, the lotus seed plant has a very adverse impact on cultivation, aquatic products, water conservancy, shipping, species resources, and the environment and has become an important foreign malignant species that needs to be eliminated [27,28]. Previous studies found that *Alternanthera philoxeroides* biochar (APBC) could remove Pb(II) efficiently, and the maximum adsorption capacity of APBC for Pb(II) was 257.12 mg/g [29]. Hydrogen-peroxide-modified biochar (mBC) derived from *Alternanthera philoxeroides* biomass was used to investigate the adsorption properties of metformin hydrochloride. The adsorption kinetics and isotherms indicated that the adsorption process of metformin hydrochloride on mBC fitted better to the pseudo-second-order model and Freundlich model, respectively [30]. According to scanning electron microscopy and pore size analyses, APBC exhibited a porous structure with a specific surface area of 857.5 m$^2$/g [31]. The low cytotoxicity of APBC demonstrated its low toxicity in potential environmental applications, so it is worth paying attention to whether APBC also has a better removal effect on antibiotics. Meanwhile, the application of APBC and modified APBC in soil antibiotics remediation needs to be further discussed. This research mainly aims to provide a reference for soil-pollution prevention and the control of antibiotics by adding a different proportion of acid- and alkali-modified APBC in purple soil.

## 2. Materials and Methods

### 2.1. Sample Collection and Processing

From June to August 2018, representative farmland within 50 m of the Jialing River basin was selected for sampling. The following three sampling sites were selected: Hechuan (HC), Jialing (JL), and Cangxi (CX), and 0–20 cm of soil samples were collected at each point. The collected soil samples were dried naturally, and the impurities, such as plant and animal residues, stones, and gravels, were removed and then passed through a 100-mesh nylon sieve. The pH, CEC, TOC content, specific surface area, and Cu content were determined. Table 1 shows the physical and chemical properties of each soil sample.

**Table 1.** Basic physical and chemical properties of different soil samples.

| Soil Sample | pH | CEC (mmol/kg) | TOC Content (g/kg) | Specific Surface Area (m$^2$/g) | Copper Content (mg/kg) |
|---|---|---|---|---|---|
| HC | 8.16 | 120.72 | 15.75 | 90.34 | 18.84 |
| JL | 7.70 | 204.08 | 12.28 | 130.21 | 16.32 |
| CX | 6.56 | 100.69 | 25.83 | 89.34 | 24.74 |

Chlortetracycline (CTC) was used as the antibiotic, was purchased from Shanghai Aladdin Biochemical Technology Co., Ltd. (Shanghai, China) and had a purity of 99.9%. Three ionizable groups exist in the molecular structure of CTC, and the ionization equilibrium constants of pKa$_1$, pKa$_2$, and pKa$_3$ are 3.60, 7.52, and 9.88, respectively. CTC with a pH < 3.60 is positively charged, with a pH between 3.60 and 7.52 is positively or negatively charged, and that with pH > 9.88 exists as an anion. Figure 1 shows the structural formulas and charge character of CTC.

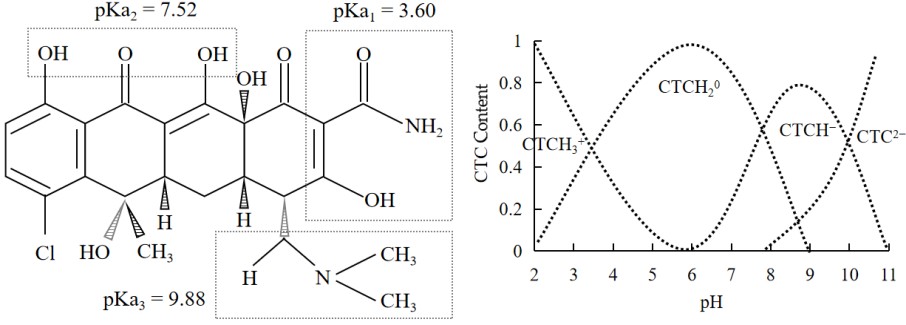

**Figure 1.** Structural formulas and charge character of CTC.

### 2.2. Preparation of Acid-Base Modified APBC

*Alternanthera philoxeroides* were from the Phase II experimental field of the China West Normal University, Sichuan Province, China. They were washed with deionized water (dH$_2$O) and natural cooling, baked to a constant weight in an oven at 60 °C, ground and reserved for use.

APBC could be obtained after placing the *Alternanthera philoxeroides* biomass in a muffle furnace at 600 °C and firing without oxygen for 4 h. The prepared APBC was soaked in HCl (1:1) for 4 h, filtered, and washed with dH$_2$O to neutralize. Then, the samples were placed in an oven at 60 °C and dried to a constant weight. After being soaked in 2 mol/L NaOH for 4 h, filtered, and washed with dH$_2$O to neutralize again, baked to constant weight, ground, and sifted through 100 mesh, acid–base modified APBC (C$_m$) was prepared. Figure 2 is presents the process of C$_m$ preparation.

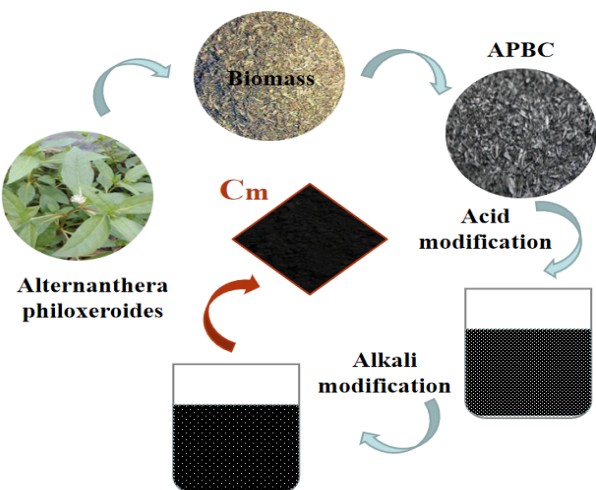

**Figure 2.** The process routing of $C_m$ preparation.

### 2.3. Preparation of Amended Soil Samples

In the following experimental design, the original soil sample (CK) was used as a control group. In this process, $C_m$ at a 2%, 5%, and 10% mass ratio were added to the CX, JL, and HC riverbank soil samples and evenly to the mixed soil samples, respectively. The Amended CX soil sample was used as an example. The soil sample mainly consisted of HC, 2% $C_m$HC (HC soil sample with 2% $C_m$), 5% $C_m$HC (HC soil sample with 5% $C_m$), and 10% $C_m$HC (HC soil sample with 10% $C_m$). CTC isotherm adsorption experiments were carried out and each treatment was repeated thrice.

### 2.4. CTC Adsorption Experiments

#### 2.4.1. CTC Concentration Gradient

CTC was set to nine concentration gradients of 0, 0.3, 1.2, 3, 6, 12, 18, 24, and 30 mg/L, with three replicates for each treatment. The temperature and pH value were 30 °C and 6, respectively.

#### 2.4.2. Influence of Environmental Factors

Experimental temperatures were set at 10 °C, 20 °C, 30 °C, and 40 °C (the pH value of the initial solution was 6, and the ionic strength of the initial solution was 0.1 mol/L); the pH value of the initial solution was set to 2, 4, 6, 8, and 10 (the initial solution temperature was 30 °C, and the ionic strength of the initial solution was 0.1 mol/L); the ionic strength of the initial solution was set to 0.01, 0.05, 0.1, 0.2 and 0.5 mol/L NaCl (the initial solution temperature was 30 °C, and the pH value of the initial solution was 6).

#### 2.4.3. Experimental Methods

A batch-equilibration method was used for CTC adsorption. In this method, 0.5000 g of the different soil samples was successively added to each 50 mL plastic plug centrifuge tube, and 20.00 mL of the CTC solution was added to each centrifuge tube. After oscillating 150 r/min for 12 h at a constant temperature, the supernatant was separated with a high-speed centrifuge at 4800 rmp for 15 min. The concentration of CTC in the supernatant was determined by HPLC. The equilibrium sorption of CTC on each test soil sample was calculated using the subtraction method.

### 2.5. Data Processing

#### 2.5.1. Calculation of Equilibrium Adsorption Amount of CTC

The equilibrium adsorption amount was calculated using Equation (1):

$$q_e = \frac{(C_0 - C_e)}{m} \tag{1}$$

where $q_e$ (mg/g) is the adsorption capacity of the $C_m$ amended soil on CTC, $V$ (mL) refers to the volume of CTC added, $C_0$ (mg/L) represents the initial addition concentration of CTC solution, $C_e$ (mg/L) denotes the concentration of CTC solution after adsorption, and $m$ (g) is the weight of the tested soil sample.

### 2.5.2. Fitting of CTC Adsorption Isotherms

Three isothermal adsorption models were selected on the basis of the adsorption isotherm trend, and the isothermal equation (Equations (2)–(4)) is as follows:

Langmuir model [32]:

$$q_e = \frac{q_m K_L C_e}{1 + K_L C_e} \tag{2}$$

Freundlich model [33]:

$$q_e = K_F C_e^{\frac{1}{n}} \tag{3}$$

Henry model [34]:

$$q_e = K_H C_e \tag{4}$$

where $q_m$ indicates the maximum adsorption amount of CTC on the different soil samples, mmol/kg; and $K_L$, $K_F$, and $K_H$ are the Langmuir, Freundlich, and Henry adsorption equilibrium constants of the CTC adsorption, which can be used to measure the affinity of adsorption.

### 2.5.3. Calculation of Thermodynamic Parameters

The parameter $K_L$ in the Langmuir model denotes the apparent adsorption constant, which is equivalent to the equilibrium constant $K$. So, the thermodynamic parameters calculated by $K$ is known as the apparent thermodynamic parameter. The calculation formulas (5)–(7) are as follows.

$$\Delta G = -RT \ln K \tag{5}$$

$$\Delta H = R \left( \frac{T_1 \cdot T_2}{T_2 - T_1} \right) \cdot \ln \left( \frac{K, T_2}{K, T_1} \right) \tag{6}$$

$$\Delta S = \frac{\Delta H - \Delta G}{T} \tag{7}$$

where $\Delta G$ is the standard free energy change (kJ/mol), $R$ is a constant (8.3145 J/mol/K), $T$ is the adsorption temperature ($T_1$ = 293.16 K, $T_2$ = 313.6 K), $\Delta H$ is the enthalpy of the adsorption process (kJ/mol), and $\Delta S$ is the entropy change of the adsorption process (J/mol/K).

The CurveExpert 1.4 fitting software (Biolead, Beijing, China) was used in isothermal fitting, and the SigmaPlot 10.0 software (StatSoft, Tulsa, OK, USA) was adopted to improve data plotting. SPSS 16.0 statistical analysis software (IBM, Armonk, NY, USA) was used to process the experimental data for variance and correlation analysis.

## 3. Results and Discussion

### 3.1. Isothermal Sorption Characteristics of CTC on Different Amended Soils

An equilibrium study can determine the maximum adsorption capacity of the adsorbent and evaluate the potential practical application value of the adsorbent [35]. Figure 3 reflects the relationship between the amount of CTC adsorbed and the initial concentration at 30 °C and pH = 6. The sorption capacity gradually increased as the initial CTC concentration rose from 0 mg/L to 30 mg/L. The sorption capacity of each soil sample increased with the equilibrium concentration. Soil samples amended with different proportions of $C_m$ showed a better equilibrium uptake of CTC than that of unamended ones. The maximum adsorption amount ($q_m$) of CTC for each tested sample had ranges of 2054.63–3631.21 mg/kg, and the adsorption capacity in different $C_m$ amended soils were ranked in the order of 10% $C_m$ > 5% $C_m$ > 2% $C_m$ > CK (Table 2). Under the same addition ratio of $C_m$, the HC soil sample had a better adsorption effect for CTC than that in JL and CX.

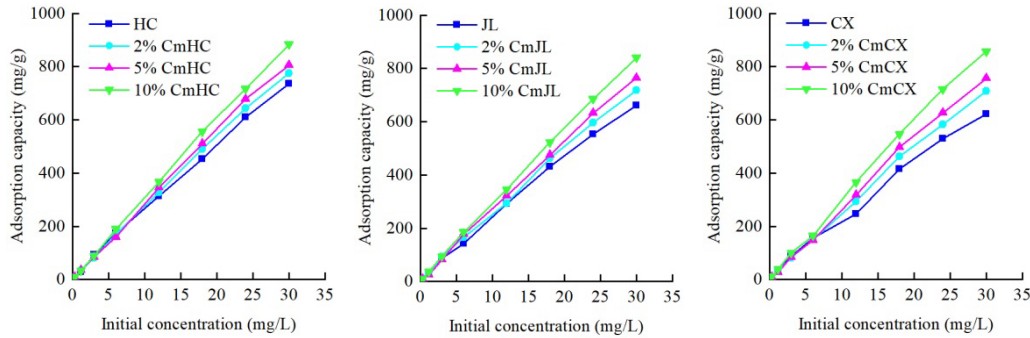

**Figure 3.** Adsorption isotherms of CTC different soil samples.

**Table 2.** Fitting parameters of CTC adsorption isotherms by three models.

| Soil Sample | Freundlich | | | Langmuir | | | Henry | |
|---|---|---|---|---|---|---|---|---|
| | $K_F$ | $1/n$ | $R^2$ | $q_m$ | $K_L$ | $R^2$ | $K_H$ | $R^2$ |
| HC | 101.1779 | 0.8124 | 0.9935 | 2482.7106 | 0.0361 | 0.9927 | 64.0721 | 0.9871 |
| 2% $C_mHC$ | 106.3882 | 0.8527 | 0.9955 | 2920.7151 | 0.0346 | 0.9984 | 75.8736 | 0.9896 |
| 5% $C_mHC$ | 114.1008 | 0.8750 | 0.9858 | 3322.1241 | 0.0338 | 0.9919 | 86.9406 | 0.9822 |
| 10% $C_mHC$ | 149.6066 | 0.8740 | 0.9941 | 3631.2122 | 0.0414 | 0.9971 | 116.7659 | 0.9895 |
| JL | 83.3789 | 0.8061 | 0.9946 | 2062.6285 | 0.0355 | 0.9967 | 50.6241 | 0.9858 |
| 2% $C_mJL$ | 95.7305 | 0.8176 | 0.9911 | 2522.3476 | 0.0335 | 0.9917 | 61.0033 | 0.9854 |
| 5% $C_mJL$ | 99.9351 | 0.8609 | 0.9930 | 3058.8482 | 0.0309 | 0.9951 | 72.3188 | 0.9884 |
| 10% $C_mJL$ | 132.8579 | 0.8436 | 0.9977 | 3375.9697 | 0.0368 | 0.9973 | 94.5804 | 0.9929 |
| CX | 77.5658 | 0.7873 | 0.9793 | 2054.6277 | 0.0307 | 0.9797 | 65.2785 | 0.9701 |
| 2% $C_mCX$ | 84.9150 | 0.8547 | 0.9918 | 2791.8392 | 0.0280 | 0.9946 | 68.8163 | 0.9790 |
| 5% $C_mCX$ | 94.5608 | 0.8803 | 0.9814 | 3157.2924 | 0.0292 | 0.9879 | 75.1827 | 0.9913 |
| 10% $C_mCX$ | 144.9343 | 0.8480 | 0.9798 | 3264.2293 | 0.0432 | 0.9859 | 87.8175 | 0.9892 |

Table 2 shows the parameters of CTC adsorption for each amended soil sample fitted by the Langmuir, Freundlich, and Henry models. Langmuir adsorption isotherm considered the process to occur by monolayer adsorption on a homogeneous surface [32]. The Freundlich model suggested the adsorption to be heterogeneous, and adsorption was carried out through a multi-layer adsorption mechanism [33]. In addition, the adsorption energy of different active sites was not equal [36]. According to the fitting correlation coefficient ($R^2$) of the isothermal adsorption equation, the Langmuir model had the best fitting effect on CTC adsorption data, with an average $R^2$ value of 0.9924. Moreover, the adsorption fitting of CTC in all soil samples had a significant correlation. The Freundlich model had a good fitting effect on the CTC adsorption isotherms, with an average $R^2$ value of 0.9898. The Henry model had the worst fitting performance, with an average $R^2$ value of 0.9859. Therefore, the Langmuir, Freundlich, and Henry models all had good results in fitting the adsorption isotherms of CTC. Nonetheless, the Langmuir model was best suited for fitting the sorption behavior of CTC on the soil. The Langmuir-fit coefficients for each soil sample were all greater than 0.9, indicating that the sorption of CTC on each amended soil sample occurred by monolayer sorption.

Original purple soil has a high specific surface area and abundant surface adsorption sites and can form a weak ion exchange effect on CTC [37]. When raw soil was amended by $C_m$, the CTC were adsorbed through negative charge points, such as the carboxyl group of $C_m$ and residual negative charge of raw soil through ion exchange. Moreover, CTC and the organic phase of $C_m$ formed hydrophobic bonds on the surface of $C_m$ amended soil, and thus the $q_m$ of CTC in $C_m$ amended soil was higher than that of raw soil [35]. Compared with the abovementioned result, the $C_m$ used in soil improvement had a high adsorption capacity for CTC and was thus considered to be a good improvement material.

### 3.2. Effect of Temperature on CTC Adsorption by Different Soil Samples

Thermodynamic research aims to better understand the effect of temperature on the adsorption mechanism of CTC in various soil samples. The variation of the CTC adsorption capacity of different soil samples with temperature is shown in Figure 4. The sorption capacity of each amended soil for CTC increased with increasing temperatures, indicating that the sorption was an endothermic form of behavior. With the increase in temperature, the adsorption process was more spontaneous because of the thermal movement of molecules, and the collision between the adsorbent and CTC was more violent at a high temperature. A previous study also showed that the adsorption capacity of TC for different amended soil samples increased with temperature, thereby showing the positive effect of increasing temperature [35].

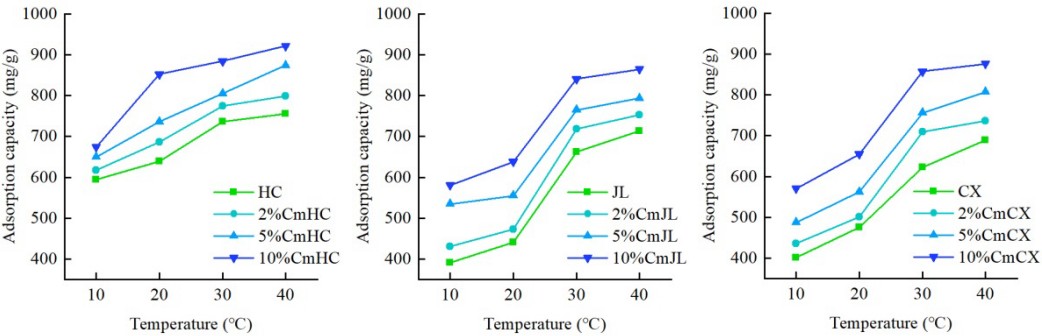

**Figure 4.** Adsorption capacity of CTC on different soil samples at various temperatures.

Table 3 shows the thermodynamic parameters of free energy ($\Delta G$), enthalpy change ($\Delta H$), and entropy change ($\Delta S$) for the CTC adsorption of each amended soil. Under the condition of 10–40 °C, the $\Delta H$ of CTC adsorption for each soil sample was greater than 0, indicating that the adsorption process was endothermic. The $\Delta G$ values in the same soil samples were $-\Delta G40 > -\Delta G10$, indicating that an increase in temperature enhanced the spontaneity of CTC adsorption. The apparent enthalpy change in the $\Delta H$ values of CTC adsorption on the tested samples were all positive, indicating that the adsorption of CTC was endothermic, and an increase in temperature enhanced adsorption. These $\Delta H$ values were completely consistent with the conclusion about temperature effect and were mutually verified. $\Delta S > 0$ indicates an increase in randomness at the solid and liquid interfaces during the adsorption process, and each soil sample has a strong affinity for CTC ion adsorption. The adsorption of antibiotics on the different tested samples was spontaneous, endothermic, and entropy increasing. The above results are consistent with those of Zou et al. [37].

**Table 3.** Thermodynamic parameters of CTC adsorption in different soil samples.

| Soil Sample | $\Delta G$ (J/mol) | $\Delta G$ (J/mol) | $\Delta G$ (J/mol) | $\Delta G$ (J/mol) | $\Delta H$ (kJ/mol) | $\Delta S$ (J/(mol·K)) |
|---|---|---|---|---|---|---|
| | 10 °C | 20 °C | 30 °C | 40 °C | | |
| HC | −2696.6208 | −2882.9380 | −3236.1411 | −3304.5867 | 12.2921 | 9.5671 |
| 2% C$_m$HC | −2559.1470 | −2826.8265 | −3129.9363 | −3205.6770 | 17.6599 | 9.1005 |
| 5% C$_m$HC | −2526.3663 | −2839.9716 | −3066.5362 | −3274.4035 | 20.6898 | 8.9954 |
| 10% C$_m$HC | −2897.4033 | −3489.4288 | −3582.5319 | −3686.2068 | 39.0583 | 10.3707 |
| JL | −1873.6183 | −2170.5904 | −3196.0392 | −3387.9110 | 19.5924 | 6.6862 |
| 2% C$_m$JL | −1750.2651 | −1986.0698 | −3044.0393 | −3159.6986 | 15.5570 | 6.2363 |
| 5% C$_m$JL | −1941.0039 | −2034.6786 | −2840.9820 | −2933.8534 | 6.1801 | 6.8769 |
| 10% C$_m$JL | −2353.2497 | −2590.8644 | −3285.9009 | −3355.7905 | 15.6764 | 8.3663 |
| CX | −1718.1698 | −2145.8742 | −2823.7779 | −3077.0638 | 28.2173 | 6.1677 |
| 2% C$_m$CX | −1368.3151 | −1723.2368 | −2596.8412 | −2691.0929 | 23.4156 | 4.9152 |
| 5% C$_m$CX | −1594.8004 | −1952.0872 | −2699.9444 | −2863.3427 | 23.5716 | 5.7156 |
| 10% C$_m$CX | −2662.5929 | −3010.3943 | −3689.7226 | −3743.7253 | 22.9458 | 9.4845 |

### 3.3. Influence of Ionic Strength on CTC Adsorption

Figure 5 shows that at pH = 6 and ionic strengths in the range of 0.01 to 0.5 mol/L, the adsorption capacity of CTC in the soil sample decreased with an increase in ionic strength. The increased amplitude of CTC adsorption decreased with the increased proportion of $C_m$. This is because the presence of NaCl may compete with CTC in the adsorption process. By competitive binding or complexation, $Na^+$ and $Cl^-$ may interfere with the soil uptake of CTC [38]. The NaCl solution may influence the adsorption of biochar through electrostatic shielding. As the ionic strength increased, the electrostatic interaction between biochar and tetracycline weakened, and the adsorption capacity of CTC decreased [39]. The soil adsorption capacity for CTC decreased with an increase in the NaCl concentration, suggesting that $Na^+$ and $Cl^-$ competed for adsorption sites in the soil. Furthermore, at high ionic strengths, the hydrophobic interactions between CTC molecules may overcome repulsive electrostatic interactions and thus facilitate aggregation. The result of this experiment was similar to the findings of other scholars about tetracycline adsorption [40,41].

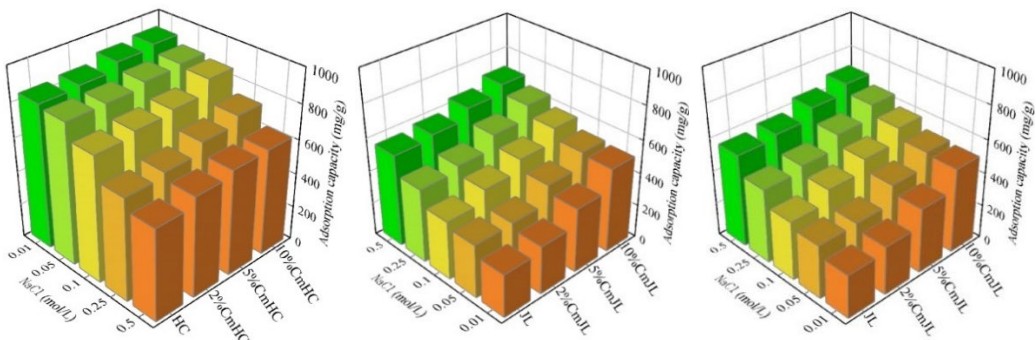

**Figure 5.** Adsorption capacity of CTC by the tested soil samples at different ionic strengths.

### 3.4. Effect of pH on CTC Adsorption by Different Soil Samples

The pH of the solution had a significant effect on the surface-functional groups, surface charges, and active sites of the adsorbents. In addition, the degree of ionization and structure of the adsorbents were affected [42]. As shown in Figure 6, with the increase in the pH value from 2 to 10, the adsorption capacity of CTC in all the amended soil samples decreased. In acidic environments, the adsorption of tetracycline by soil samples was significantly greater than that in alkaline environments. Under low pH conditions, the adsorption capacity of CTC was significantly higher than that of unmodified ones because the modified biochar provided more adsorption sites for the adsorption of tetracycline, which was more conducive to adsorption [43]. In addition, antibiotics mainly exist in the form of cations under acidic conditions. The cation exchange between CTC and $C_m$-amended soil increased.

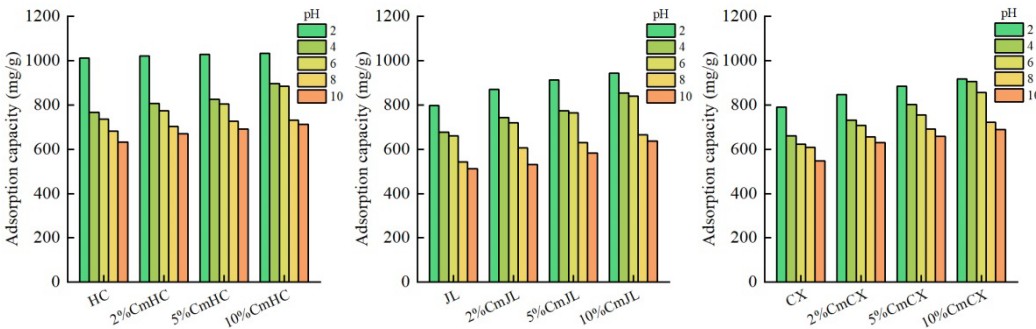

**Figure 6.** Adsorption capacity of CTC by the tested soil samples at different pH.

## 4. Conclusions

The addition of $C_m$ to soil samples led to better equilibrium absorption of CTC, showing the following trend: 10% $C_m$ > 5% $C_m$ > 2% $C_m$ > CK. Under the same addition ratio of $C_m$, the HC soil sample had a better adsorption effect for CTC. The Langmuir model was the most suitable model to fit the sorption behavior of CTC on amended soil. The thermodynamic parameters indicated that the sorption of CTC on the modified soil samples was a spontaneous, heat-absorbing, and entropy-increasing process. When the ionic strength ranged from 0.01 to 0.5 mol/L, the adsorption capacity of CTC by the soil samples decreased with an increase in the ionic concentration. In the range of pH 2–10, the adsorption capacity of CTC by the tested soil samples decreased with an increase in the pH value.

**Author Contributions:** Conceptualization, Z.L. and X.F.; methodology, L.C. and B.T.; software, F.S.; validation, Z.L. and X.F.; resources, W.L.; writing—original draft preparation, Z.L.; writing—review and editing, W.L.; project administration, W.L.; funding acquisition, Z.L. and W.L. All authors have read and agreed to the published version of the manuscript.

**Funding:** The authors wish to acknowledge and thank the Scientific Research Foundation of State Key Laboratory of Qinba Bio-Resource and Ecological Environment (SXC-2105), the Fundamental Research Funds of Shaanxi University of Technology (SLGRCQD2027) and the Scientific Research Foundation of the Education Department of Shaanxi Province (20JY008), and the financial assistance from the Scientific Research Foundation of Sichuan Science and Technology Agency (2018JY0224).

**Data Availability Statement:** Not applicable.

**Conflicts of Interest:** The authors declare no conflict of interest.

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
