# Peer review of "Effect of Acid–Base Modified Biochar on Chlortetracycline Adsorption by Purple Soil"

_sustainability, doi:10.3390/su14105892_

Round 1

Reviewer 1 Report

Totally the present article is not well-established, but the subject is quite interesting.Therefore, there is still room for narrative, argumentative, and verification improvements, prior it to be accepted for publication.

  1. Keywords: It is good if authors improve them. The function of keywords is to supplement the information given in the title. Words in the title are automatically included in indexes, and keywords serve as additional pointers.
  2. The highlight points have to be accompanied by numerical data or quantitative information, to support the outcomes yielded. The sole notation of expressions like: acceptable, improved, well-established, can meet the necessity for, they have to be quantified.
  3. The manuscript needs language, grammar and syntactic editing. The English language usage should be checked by a fluent English speaker. Grammar and syntax of narrative can be checked and smoothened, accordingly.
  4. More suitable title should be selected for the article. Title should decrease to 10-12 words.
  5. The abstract should state briefly the purpose of the research, the principal results and major conclusions. An abstract is often presented separately from the article, so it must be able to stand alone.This section isn't clear. Authors just collecting some ideas. Please, try to improve this section by highlighting the research gape and the novelty of this work. Also, try to lead the author smoothly to your point.
  6. It is suggested to present the structure of the article at the end of the introduction.
  7. The necessity and innovation of the article should be presented to the introduction.
  8. The text content of the Introduction section can be reorganized into two or three paragraphs.
  9. The major defect of this study is the debate or Argument is not clear stated in the introduction session. Hence, the contribution is weak in this manuscript. I would suggest the author to enhance your theoretical discussion and arrives your debate or argument.
  10. Please avoid reference overkill/run-on, i.e. do not use more than 3 references per sentence.
  11. A flowchart should be added to the article to show the research methodology.Authors followed a scientific and acceptable approaches; however, they fail in presenting their steps in a clear way. I recommend a second look to this section and deleting unnecessary details.
  12. It is important to cite all equations into the main text.
  13. It is suggested to compare the results of the present research with some similar studies which is done before.
  14. Much more explanations and interpretations must be added for the Results, which are not enough.
  15. The Results and Discussion section is devoted, in large, by representing the research outcomes' yielded, but a critical and integrated approach of these outcomes has been made, probable at a distinct "synthesis' and cross-cited subsection. In this distinct subsection the key-aspects that determine the outcomes have to be signified into a descriptive manner.
  16. Please make sure your conclusions' section underscore the scientific value added of your paper, and/or the applicability of your findings/results, as indicated previously. Please revise your conclusion part into more details. Basically, you should enhance your contributions, limitations, underscore the scientific value added of your paper, and/or the applicability of your findings/results and future study in this session.
  17. "Notation" should be added to the article.
  18. The citing information at the References section is given in a messy way, since it seems that the Vol and No of issues is partially missing, while page range is noted either in "pp" or not. Therefore, unification of all citations has to be given, making them easily traceable by the readers. Moreover, literature refresh and enrichment with more and relevant to the topic published papers can be deployed.
  19. Add some of the following references

---Das, S.K., Ghosh, G,K., Avasthe, R.K., Sinha, K., 2021. Compositional heterogeneity of different biochar: Effect of pyrolysis temperature and feedstocks. Journal of Environmental Management. 278 (2): 111501. https://doi.org/10.1016/j.jenvman.2020.111501

---Das, S.K., Ghosh, G,K., Avasthe, R.K., Sinha, K., 2020. Morpho-mineralogical exploration of crop, weed and tree derived biochar. Journal of Hazardous Materials. 124370. https://doi.org/10.1016/j.jhazmat.2020.124370

Author Response

Dear Professor,

Thank you for your useful comments and suggestions. We have learned much from the reviewers’ comments, which are fair, encouraging and constructive. After thinking about the comments carefully, we have revised them (red letters in the manuscript) and detailed corrections are listed (blue letters) as below:

Reviewer #1: 

  1. Keywords: It is good if authors improve them. The function of keywords is to supplement the information given in the title. Words in the title are automatically included in indexes, and keywords serve as additional pointers.

Thank you for your constructive comments in this paper. We have added more information in keywords part. Such as Alternanthera philoxeroides, modification and Jialing River.

  1. The highlight points have to be accompanied by numerical data or quantitative information, to support the outcomes yielded. The sole notation of expressions like: acceptable, improved, well-established, can meet the necessity for, they have to be quantified.

Thank you for your suggestion. We have added the core conclusion of this article and some more precise data results in abstract.

  1. The manuscript needs language, grammar and syntactic editing. The English language usage should be checked by a fluent English speaker. Grammar and syntax of narrative can be checked and smoothened, accordingly.

We have carefully proofread and redescribed the unclear description in the whole text, and tried to avoid any grammar or syntax error with the help of a native speaker (Ken Galloway).

  1. More suitable titleshould be selected for the article. Title should decrease to 10-12 words.

We have modified the title according to the comments, to ensure that the number of words is less than 12.  

  1. The abstract should state briefly the purpose of the research, the principal results and major conclusions. An abstract is often presented separately from the article, so it must be able to stand alone.This section isn't clear. Authors just collecting some ideas. Please, try to improve this section by highlighting the research gape and the novelty of this work. Also, try to lead the author smoothly to your point.

Thank you for your suggestion. We modify the parameters in the following ways: background, objects, methods, results, discussion and conclusion. Also, We have added the core conclusion of this article and some more precise data results in abstract.

  1. It is suggested to present the structure of the article at the end of the introduction.

We have added the the structure of the article at the end of the introduction.

  1. The necessity and innovation of the article should be presented to the introduction.

Thank you for your comments in this paper. In the introduction part, we added the description about the novelty and significance. If the modified biochar was added to the actual contaminated soil sample as a modified material, the adsorption capacity of improved soil for antibiotics could be greatly improved. Meanwhile, the biochar material improved the physical and chemical properties of soil simultaneously, showing a good application prospect.

  1. The text content of the Introduction section can be reorganized into two or three paragraphs.

    We have reorganized text content of the Introduction section into three paragraphs.

  1. The major defect of this study is the debate or Argument is not clear stated in the introduction session. Hence, the contribution is weak in this manuscript. I would suggest the author to enhance your theoretical discussion and arrives your debate or argument.

In the introduction part, we added the description about the novelty and significance. At the same time, we supplement a large number of literature and review the application of biochar modified materials.  

  1. Please avoid reference overkill/run-on, i.e. do not use more than 3 references per sentence.

We have used more of the recent and relevant references in the paper, and changed all references one by one to meet the right style of the references.

  1. A flowchart should be added to the article to show the research methodology.Authors followed a scientific and acceptable approaches; however, they fail in presenting their steps in a clear way. I recommend a second look to this section and deleting unnecessary details.

We have added the flowchart about the research methodology, which shown in Fig. 1.

  1. It is important to cite allequations into the main text.

We have cited the relevant reference of all equations into the main text, and it could be found in references list.

  1. It is suggested to compare the results of the present research with some similar studies which is done before.

We have added the similar researches of the adsorption amount and compare the results of previous studies, which showed in page 8-10.

  1. Much more explanations and interpretations must be added for the Results, which are not enough.

We have discussed more about the adsorption mechanism of materials and the adsorption mechanism after the addition of soils. Original purple soil has a high specific surface area and abundant surface adsorption sites and can form a weak ion exchange effect on CTC [37]. When raw soil was amended by Cm, the CTC were adsorbed through negative charge points, such as the carboxyl group of Cm and residual negative charge of raw soil through ion exchange. Moreover, CTC and the organic phase of Cm formed hydrophobic bonds on the surface of Cm amended soil, and thus the qm of CTC in Cm amended soil were higher than that of raw soil [35]. Compared with the above result, the Cm used in soil improvement had a high adsorption capacity for CTC and is thus a good improvement material.

  1. The Results and Discussion section is devoted, in large, by representing the research outcomes' yielded, but a critical and integrated approach of these outcomes has been made, probable at a distinct "synthesis' and cross-cited subsection. In this distinct subsection the key-aspects that determine the outcomes have to be signified into a descriptive manner.

We have carried out more in-depth discussions on the results and discussion sections, such as comparative studies, mechanism analysis and so on.  

  1. Please make sure your conclusions' section underscore the scientific value added of your paper, and/or the applicability of your findings/results, as indicated previously. Please revise your conclusion part into more details. Basically, you should enhance your contributions, limitations, underscore the scientific value added of your paper, and/or the applicability of your findings/results and future study in this session.

We have rewritten the conclusion section to make sure that the content is the core conclusion and the conclusion has important value.  

  1. "Notation" should be added to the article.

To make it easier to understand, we have revised the format of the full text according to the journal introduction. Then we defined at first mention of Abbreviations and used consistently thereafter.

  1. The citing information at the References section is given in a messy way, since it seems that the Vol and No of issues is partially missing, while page range is noted either in "pp" or not. Therefore, unification of all citations has to be given, making them easily traceable by the readers. Moreover, literature refresh and enrichment with more and relevant to the topic published papers can be deployed.

We have revised the format of reference numbers according to the journal introduction.

  1. Add some of the following references: ---Das, S.K., Ghosh, G,K., Avasthe, R.K., Sinha, K., 2021. Compositional heterogeneity of different biochar: Effect of pyrolysis temperature and feedstocks. Journal of Environmental Management. 278 (2): 111501. https://doi.org/10.1016/j.jenvman.2020.111501;---Das, S.K., Ghosh, G,K., Avasthe, R.K., Sinha, K., 2020. Morpho-mineralogical exploration of crop, weed and tree derived biochar. Journal of Hazardous Materials. 124370.https://doi.org/10.1016/j.jhazmat.2020.124370

We have cited this two relevant reference in the introduction section, and it could be found in references list.

Reviewer 2 Report

The article presents very important topics about adsorption effect of different proportions of acid–base modified biochar on chlortetracycline by purple soil. The article is very interesting, the methodology planned correctly. Unfortunately the authors did not avoid mistakes. Below is a list of things to fix:

In the introduction, there is no comparison to other adsorption methods, the article to be cited in brackets (https://doi.org/10.12911/22998993/74272 and http://dx.doi.org/10.5004/dwt.2018.22551)

The materials and methods lack a process diagram or a photo of the biochar with a detailed description.

in verse 41, 42 and 45 are mistake in citation. You should check the entire article in this direction.

Why was only the adsorption of  chlortetracycline  studied?

Why was the BET isotherm not used?

The Discussion of results section should be added, where the results of the adsorption of   chlortetracycline will be compared with the adsorption on other materials.

Author Response

Dear Professor,

Thank you for your useful comments and suggestions. We have learned much from the reviewers’ comments, which are fair, encouraging and constructive. After thinking about the comments carefully, we have revised them (red letters in the manuscript) and detailed corrections are listed (blue letters) as below:

Reviewer #2: 

  1. In the introduction, there is no comparison to other adsorption methods, the article to be cited in brackets (https://doi.org/10.12911/22998993/74272 and http://dx.doi.org/10.5004/dwt.2018.22551)

We have cited the relevant reference (http://dx.doi.org/10.5004/dwt.2018.22551) in the introduction section, and it could be found in references list.

  1. The materials and methods lack a process diagram or a photo of the biochar with a detailed description.

Thank you for your constructive comments in this paper. We have added the process diagram about the materials, which shown in Fig. 1.

  1. in verse 41, 42 and 45 are mistake in citation. You should check the entire article in this direction.

We have rewritten the revised part to ensure that the language structure is reasonable, without grammatical errors, and is consistent.  

  1. Why was only the adsorption ofchlortetracycline studied?

Oxytetracycline is the most toxic tetracycline antibiotics and the most harmful to the environment, so we choose oxytetracycline.  

  1. Why was the BET isotherm not used?

Based on the trend of adsorption isotherms, we choose a more similar model for fitting.  

  1. The Discussion of results section should be added, where the results of the adsorption ofchlortetracycline will be compared with the adsorption on other materials.

We have added the similar researches of the adsorption amount and compare the results of previous studies, which showed in page 8-10.

Reviewer 3 Report

  1. The novelty of the present study is needed to be highlighted.
  2. What is the difference between the present study and the published manuscript "Zou, Y., Deng, H. Y., Li, M., Zhao, Y. H., & Li, W. B. (2020). Enhancing tetracycline adsorption by riverbank soils by application of biochar-based composite materials. Desalination and Water Treatment207(2), 332-340."
  3. Subsections under section 2.4 needed to be rewritten properly.
  4. Conclusion should be rewritten properly. 
  5. Overall the manuscript lacks novelty2.

Author Response

Dear Professor,

Thank you for your useful comments and suggestions. We have learned much from the reviewers’ comments, which are fair, encouraging and constructive. After thinking about the comments carefully, we have revised them (red letters in the manuscript) and detailed corrections are listed (blue letters) as below:

Reviewer #3: 
1. The novelty of the present study is needed to be highlighted.

Thank you for your comments in this paper. In the introduction part, we added the description about the novelty and significance. If the modified biochar was added to the actual contaminated soil sample as a modified material, the adsorption capacity of improved soil for antibiotics could be greatly improved. Meanwhile, the biochar material improved the physical and chemical properties of soil simultaneously, showing a good application prospect.

  1. What is the difference between the present study and the published manuscript "Zou, Y., Deng, H. Y., Li, M., Zhao, Y. H., & Li, W. B. (2020). Enhancing tetracycline adsorption by riverbank soils by application of biochar-based composite materials. Desalination and Water Treatment, 207(2), 332-340."

This paper focuses on the effects of biochar modification with different ratios of acid and base, and this paper mainly focuses on several applications of biochar-based materials.  At the same time, the target of pollutants is different.  

  1. Subsections under section 2.4 needed to be rewritten properly.

We have rewritten the revised part to ensure that the language structure is reasonable, without grammatical errors, and is consistent.  

  1. Conclusion should be rewritten properly. 

We have rewritten the conclusion section to make sure that the content is the core conclusion and the conclusion has important value.  

Round 2

Reviewer 2 Report

The article after thorough corrections is suitable for publication. The authors took into account all the comments, corrected and improved the manuscript.

Reviewer 3 Report

The manuscript lacks novelty.

This manuscript is a resubmission of an earlier submission. The following is a list of the peer review reports and author responses from that submission.